# Effectiveness of a Nanohydroxyapatite-Based Hydrogel on Alveolar Bone Regeneration in Post-Extraction Sockets of Dogs with Naturally Occurring Periodontitis

**DOI:** 10.3390/vetsci9010007

**Published:** 2021-12-26

**Authors:** Kittidaj Tanongpitchayes, Chamnan Randorn, Suphatchaya Lamkhao, Komsanti Chokethawai, Gobwute Rujijanagul, Kannika Na Lampang, Luddawon Somrup, Chavalit Boonyapakorn, Kriangkrai Thongkorn

**Affiliations:** 1Master’s Degree Program in Veterinary Science, Faculty of Veterinary Medicine, Chiang Mai University, Chiang Mai 50100, Thailand; kittidaj_tanong@cmu.ac.th; 2Small Animal Hospital, Faculty of Veterinary Medicine, Chiang Mai University, Chiang Mai 50200, Thailand; luddawan.s@cmu.ac.th (L.S.); chavalit.b@cmu.ac.th (C.B.); 3Innovation Center for Holistic Health, Nutraceuticals, and Cosmeceuticals, Faculty of Pharmacy, Chiang Mai University, Chiang Mai 50200, Thailand; 4Department of Chemistry, Faculty of Science, Chiang Mai University, Chiang Mai 50200, Thailand; chamnan.r@cmu.ac.th (C.R.); suphatchaya_l@cmu.ac.th (S.L.); 5Department of Physics and Materials Science, Faculty of Science, Chiang Mai University, Chiang Mai 50200, Thailand; komsanti.chokethawai@cmu.ac.th (K.C.); gobwute.ruji@cmu.ac.th (G.R.); 6Department of Veterinary Bioscience and Veterinary Public Health, Faculty of Veterinary Medicine, Chiang Mai University, Chiang Mai 50100, Thailand; kannika.nalampang@cmu.ac.th; 7Department of Companion Animals and Wildlife Clinic, Faculty of Veterinary Medicine, Chiang Mai University, Chiang Mai 50100, Thailand; 8Integrative Research Center for Veterinary Circulatory Sciences, Faculty of Veterinary Medicine, Chiang Mai University, Chiang Mai 50100, Thailand

**Keywords:** veterinary dentistry, veterinary periodontology, nanohydroxyapatite, hydrogel, alveolar bone regeneration, periodontitis, dogs

## Abstract

Pathological mandibular fracture after dental extraction usually occurs in dogs with moderate to severe periodontitis. A nanohydroxyapatite-based hydrogel (HAP hydrogel) was developed to diminish the limitations of hydroxyapatite for post-extraction socket preservation (PSP). However, the effect of the HAP hydrogel in dogs has still not been widely investigated. Moreover, there are few studies on PSP in dogs suffering from clinical periodontitis. The purpose of this study was to evaluate the effectiveness of the HAP hydrogel for PSP in dogs with periodontitis. In five dogs with periodontitis, the first molar (309 and 409) of each hemimandible was extracted. Consequently, all the ten sockets were filled with HAP-hydrogel. Intraoral radiography was performed on the day of operation and 2, 4, 8 and 12 weeks post operation. The Kruskal–Wallis test and paired *t*-test were adopted for alveolar bone regeneration analysis. The results demonstrated that the radiographic grading, bone height measurement, and bone regeneration analysis were positively significant at all follow-up times compared to the day of operation. Moreover, the scanning electron microscopy with energy-dispersive X-ray spectroscopy imaging after immersion showed a homogeneous distribution of apatite formation on the hydrogel surface. Our investigation suggested that the HAP hydrogel effectively enhances socket regeneration in dogs with periodontitis and can be applied as a bone substitute for PSP in veterinary dentistry.

## 1. Introduction

Today, owners’ knowledge of animal care can promote the longevity of dogs. As a result, periodontal disease (PD) is one of the health problems suffered by senile dogs [1]. This problem can occur in dogs over three years of age, with the disease having a reported prevalence of more than 80% and usually arising in small-breed dogs [2,3]. PD is the result of the inflammatory response of the periodontal apparatus to dental plaque on the tooth surface [4]. The American Veterinary Dental College classifies periodontitis into four stages according to the degree of alveolar bone destruction. It is suggested that in moderate (stage 3) and advanced periodontitis (stage 4) cases, where there is alveolar bone loss of more than 50%, dental extraction should be performed. The strength of the post-extraction socket is diminished due to severe vertical and horizontal alveolar bone loss [5]. This is the reason why pathological fractures after dental extraction are reported in dogs with moderate and advanced periodontitis [6,7]. Previous studies have reported that the incidence of pathological fracture in the mandible in dogs is related to the first mandibular molar teeth [7,8,9]. This location usually shows moderate and advanced periodontitis more than other sites [10]. To prevent pathological fracture after extraction, tooth extraction with an atraumatic technique or additional treatments such as bone grafting at the post-extraction socket may reduce the risk of pathological fracture after dental extraction.

Post-extraction socket preservation (PSP) is a widespread technique in human dentistry. This technique is commonly applied in the case of planning for future dental implant placement. As alveolar bone resorption can happen spontaneously after dental extraction [11,12], this technique can minimize alveolar bone resorption and enhance alveolar bone regeneration by applying bone graft material to the post-extraction socket. To date, many studies in humans have reported the success of PSP in different materials, such as demineralized freeze-dried bone allograft [13], bovine bone mineral [14], hydroxyapatite and beta-tricalcium [15]. On the contrary, there are only a few studies on bone graft material for PSP in dogs. Moreover, previous studies have often reported the achievement of PSP in experimental animals. Thus, investigating post-extraction socket preservation in dogs with periodontal disease is still crucial to fulfilling clinical practice in veterinary dentistry.

Hydroxyapatite (Ca_5_(PO_4_)_3_(OH)_2_) is an alloplastic material commonly used in orthopedic and dental surgery. This material consists of a calcium and phosphate-based (Ca-P) substrate with a Ca/P ratio of 1.67; its particle size is approximately 10–500 nm, and it has better biocompatibility than other groups [16]. HA has osteoconduction properties which provide a scaffold to enhance angiogenesis and osteogenesis at grafting areas. Consequently, this material is known to promote bone regeneration through its application as a bone graft material or in combination with medical implants such as a coating on the bone implant [17], prosthetic implant [18], synthetic mesh [19], or post-extraction socket preservation [15,20]. Previous studies have reported that nanohydroxyapatite, which has a small particle size of approximately 20–50 nm, showed excellent biocompatibility, with better osteoconduction properties reported than traditional hydroxyapatite [21]. Moreover, its nanoparticle size characteristics can provide a larger surface area to promote osteoconductive properties [22]. Thus, nanohydroxyapatite has become an interesting material for PSP, and was investigated in this study. However, the characteristics of hydroxyapatite material include hardness, fragility, and perhaps difficulty in organizing an appropriate shape. In addition, the characteristics of hydroxyapatite distribution within the socket may affect socket healing processes. Therefore, a hydroxyapatite combination with a hydrogel can serve as a scaffold of graft material that may erase the limitation of formulating the material in the post-extraction socket. 

A hydrogel is a hydrophilic polymer that can absorb and retain water or biological fluid without losing its structure [23]. This material is usually applied in biomedical engineering and for drug delivery [24], dental implants [25,26] and tissue engineering [27,28]. Many studies have demonstrated that hydrogels have osteoconductive properties, because they have a three-dimensional structure and provide a scaffold structure for osteoblast infiltration and neovascularization from the host’s tissue [29,30]. Moreover, hydrogels have good biocompatibility and degradability. These characteristics are critical for the bone regeneration process, because the nonabsorbable or inappropriate degradation time of the material can interfere with the alveolar bone healing process [31]. However, while many studies on hydroxyapatite composited with hydrogels for alveolar bone augmentation have been reported in human dentistry, the development of a hydroxyapatite-composited hydrogel for post-extraction socket preservation in veterinary dentistry is still lacking. A nanohydroxyapatite-based hydrogel (HAP hydrogel), which is made up of nanohydroxyapatite with a polyacrylamide hydrogel, was prepared in our research. This material promises to diminish the limitations of hydroxyapatite. In addition, the distribution of graft material in the hydrogel scaffold may facilitate cell infiltration, and tends to ensure more biocompatibility for alveolar bone augmentation. 

To evaluate the effectiveness of the nanohydroxyapatite-based hydrogel in promoting alveolar bone regeneration in the post-extraction sockets of dogs with naturally occurring periodontitis, this investigation assessed the potential of HAP hydrogel biocompatibility by scanning electron microscopy (SEM) and energy-dispersive X-ray spectroscopy (EDS) mapping analysis, together with material immersion in a simulated body fluid (SBF). Moreover, post-extraction socket preservation with the nanohydroxyapatite-based hydrogel was evaluated in dogs with periodontitis by assessing the characteristics of intraoral radiographic finding and grading, changing the alveolar socket height, and altering the socket radiodensity 2 weeks, 4 weeks, 8 weeks and 12 weeks postoperatively. Further, it was proven that the nanohydroxyapatite-based hydrogel is a promising candidate for enhancing alveolar bone regeneration, and that it also might decelerate tooth resorption of the post-extraction sockets of dogs with periodontitis. In addition, this investigation featured biomedical engineering development, provides effective material and easy accessibility, and serves as a practical choice for the management of periodontal disease in dogs. 

## 2. Materials and Methods

### 2.1. Animal and Sample Selection

A total of five small-breed dogs (ten sockets) with age >3 years regardless of sex and body weight that were brought in as outpatients to the dental clinic, Small Animal Teaching Hospital, the Faculty of Veterinary Medicine, Chiang Mai University (FVM-CMU) with oral problems and suspected periodontal disease were recruited for this study. The appropriate number of tested sockets was calculated from the difference of means within the group. The mean and standard deviation of new alveolar bone formation before and after grafted material placement were 45.95 ± 4.41 and 50.10 ± 5.01, respectively. These values were adopted from Ho’s study [32]. The α error probability and power (1-β probability of error) were 0.05 and 0.8, respectively. The sample size was calculated using G*Power 3.1.9.7. (G*Power Software, Dusseldorf, Germany). Finally, the total number of sockets for assessing the effectiveness of the HAP hydrogel was ten sockets. Physical examination and pre-anesthetic assessment, which consisted of a complete blood count profile, blood chemistry profile, and thoracic radiography, were performed before the surgical operation to assess American Society of Anesthesiologists Physical Status Classification (ASA Status). Dogs with an ASA Status level higher than ASA III (Appendix A), an uncontrollable systemic disease, or that were taking medication that alters calcium or phosphate absorption or metabolism were excluded. The protocol for the use of animals was approved by the Animal Care and Use Committee, Faculty of Veterinary Medicine, Chiang Mai University, Thailand (FVM-CMU-ICUC Ref. No. S2/2563). 

For all of the dogs that fulfilled the inclusion criteria, an anesthetic consent form was required from their owners before they received general anesthesia. Then, the dogs received general anesthesia using the general anesthesia protocol in the surgery unit, small animal teaching hospital, FVM-CMU (Appendix A), and the processes of general anesthesia were performed by an experienced veterinary anesthesiologist. Periodontal disease assessment, including attachment loss evaluation and periodontal depth probing, was performed for all dogs by one experienced veterinary dentist. The stage of periodontal disease (Appendix A) was classified for the recruited dogs using their mandibular first molar teeth to confirm that they were categorized into periodontal disease grades 3 or grade 4.

### 2.2. Preparation of Nanohydroxyapatite-Based Hydrogel (HAP Hydrogel)

The HAP hydrogel, a combination of a polyacrylamide-based hydrogel and nanohydroxyapatite, was prepared by the Department of Chemistry, Faculty of Science, Chiang Mai University. Then, the HAP hydrogel was contained in a 0.5 mL Eppendorf tube and sterilized via a low-temperature plasma sterilizer with a temperature below 50 °C before it was used for post-extraction socket preservation. The cytotoxic test of synthetic nanohydroxyapatite was performed following the study of Lamkhao et al., 2019 [33], and it was found that this material has no cytotoxic effect on cell viability from lactate dehydrogenase measurement.

### 2.3. The Characteristics of HAP hydrogel and Immersion Results in Simulated Body Fluid (SBF)

The biocompatibility of the HAP hydrogel was assessed by immersing the material in a simulated body fluid (SBF), which is also referred to as human blood plasma with ionic concentration, and observing the new apatite formation on the material’s surface. The SBF in this study was processed by following Kokubo’s protocol [34]. In addition, scanning electron microscopy (SEM) with energy-dispersive X-ray spectroscopy (EDS) analysis was performed using a JEOL JSM-6335 F electron microscope (Jeol JSM-6335 F, JEOL Ltd., Tokyo, Japan) to evaluate the distribution of hydroxyapatite and apatite formation on the hydrogel’s surface before and after immersion with the SBF for 14 days. Moreover, SEM mapping and SEM image corresponding elemental mapping were also carried out to assess the presence of interesting elements in the HAP hydrogel.

### 2.4. Post-Extraction Socket Preservation with HAP Hydrogel

Before the initial operation, mouth washing with 0.12% chlorhexidine gluconate (C-20^®^, Osoth Inter Lab, Bangkok, Thailand) and professional dental scaling and polishing were performed. Unilateral mandibular first molar teeth of dogs with stage 3 or 4 periodontitis were selected for post-extraction socket preservation. Further, 2% lidocaine hydrochloride (Locana^®^) was applied for local anesthesia at the inferior alveolar nerve block before extraction. An open dental extraction technique was used, and the closing of the post-extraction socket with the mucoperiosteal flap was carried out. Only one board-certified veterinary surgeon performed the surgical procedure in this study. 

Ten post extraction sockets (*n* = 10) from five dogs were selected to be filled with the HAP hydrogel locally (Figure 1). The post-extraction sockets were closed with the mucoperiosteal flap using poliglecaprone 23 (Monocryl^®^) 4/0 with a simple interrupted suture pattern. Postoperative care was carried out by administering carprofen (Rimadyl^®^) once daily in 4.4 mg/kg doses for anti-inflammation, and analgesia for 4 days and amoxicillin-clavulanic acid (Cavumox^®^) in 15 mg/kg doses was given twice daily for a postoperative antibiotic.

### 2.5. Intra-Oral Dental Radiography

A Genoray PORT X II (Genoray^®^, Seongnam City, Korea) portable dental radiographic system was used for intraoral radiography with the parallel technique on the first mandibular molar tooth. The film focal distance was approximately 3 to 5 cm, and the suggested exposure settings include 60–70 kV and 20–25 mA for small dog breeds [35]. Each dog was admitted for intraoral radiography on the day of the operation (Ex-0). Then, alveolar bone regeneration follow-up at the post-extraction socket sites was also performed 2 weeks (Ex-2), 4 weeks (Ex-4), 8 weeks (Ex-8), and 12 weeks after extraction (Ex-12). All of the radiographs were collected with the Vet Exam plus 9.6.0. program. To evaluate in the weeks after extraction the effectiveness of the nanohydroxyapatite-based hydrogel in promoting post-extraction socket preservation, alveolar bone regeneration analysis, which includes radiographic evaluation and grading, bone height analysis, and bone density analysis from these radiographs were performed.

### 2.6. Radiographic Evaluation and Grading

All radiographs were interpreted by three well-trained examiners using criteria of alveolar socket regeneration developed for the study. The radiographic findings of post-extraction socket preservation in the weeks of the follow-up were described. Moreover, these criteria, which were adapted from previous studies [36,37,38], and which included characteristics of the surgical site margin, as seen in Figure 2 (Criteria A), and characteristics of alveolar new bone formation, as seen in Figure 3 (Criteria B), were applied for post-extraction socket grading. Two of the three examiners had to agree before each socket grade was accepted in order to prevent bias from radiographic evaluation.

### 2.7. New Alveolar Bone Height Measurement

New alveolar bone height measurement in this study was adapted from the study of Sun [20], and all of the reference lines are shown in Figure 4. First of all, the S line was drawn from the mesial and distal crest of the grafted socket to the dorsal part of the compact bone. These lines were noted as the S1 and S2 lines. The midpoint of the S1 line and the S2 line at the coronal part and apical were defined as the C(s)1 and C(s)2 points, respectively. In addition, the thickness of the compact bone at the S1 and S2 lines, like that at the C1 and C2 lines, was drawn to be the reference thickness when making comparisons across different follow-up weeks. 

When new alveolar bone height measurement was performed, the white dash line, which was drawn to start at the C(s)1 point and continue to the S point, was noted as the reference line for measurement. The distance of the overlapping reference line from C(s)2 to the highest part of the radiopaque area was incorporated into the new alveolar bone height. This variable was analyzed to assess alterations in new alveolar bone height (Av.Bh) within the socket in different follow-up weeks.

### 2.8. Alveolar Bone Density Analysis

The bone density of the post-extraction sockets, which indicates the amount of alveolar bone regeneration, was assessed using the Image J program (NIH, Bethesda, MD, USA). All radiographs, which used the red-green-blue format (RGB), were converted to grayscale (black/white) images before pixel value evaluation. After this, the total pixel values of the sockets were calculated. Then, conversion of the pixel values to the intensity, which refers to the density of alveolar bone regeneration in the sockets, was performed. The range of intensity was between 0 to 255, with 255 indicating the most radiopacity and 0 indicating the most radiolucent area in each socket. The mean of the alveolar socket intensity (Av.Bi) was reported to refer to as the alveolar bone density in each follow-up week.

### 2.9. Statistical Analysis

Statistical analysis was performed with RStudio Software 1.4.1717 (RStudio, Inc., Boston, MA, USA). A value of *p* < 0.05 was determined to be statistically significant. The signalments of the recruited dogs, including age, sex, breed, body weight, periodontal stages, and concurrent disease, are presented as descriptive statistics. The data from alveolar bone regeneration analysis, including the radiograph grading, new alveolar bone height, and alveolar bone intensity, were subjected to D’Agostino and Pearson’s normality test prior to performing statistical analysis. To evaluate the effectiveness of the nanohydroxyapatite-based hydrogel in promoting alveolar bone regeneration in the post-extraction sockets of the recruited dogs, a paired *t*-test was used to compare differences between the mean alveolar bone height and mean alveolar bone intensity of the sockets for Ex-0 and Ex-2, Ex-4, Ex-8, and Ex-12. Moreover, the Kruskal–Wallis test (nonparametric test) was used to describe the differences in radiograph grading in different weeks of follow-up. Data from the statistical analysis are represented as graphs generated by GraphPad Prism 9 (GraphPad Software, San Diego, CA, USA) and RStudio Software.

## 3. Results

### 3.1. Animals

The signalments of the recruited dogs are summarized in Table 1. Unilateral first mandibular molar extraction was carried out in each dog, and two roots of the first mandibular molar teeth were noted as two post-extraction sockets. Therefore, the study performed HAP hydrogel placement on ten post-extraction sockets (*n* = 10) from five dogs with periodontitis grades 3 and 4. Unfortunately, three sockets (*n* = 3) were excluded from the study two weeks post-operation because one pathological mandibular fracture and two mucoperiosteal flap dehiscences were found. However, the rest of the post-extraction sockets (*n* = 7) were followed-up completely to twelve weeks post-operation.

### 3.2. Scanning Electron Microscopy (SEM) and EDS Mapping Analysis of Nanohydroxyapatite-Based Hydrogel and Immersion Results in Simulated Body Fluid (SBF) for 14 Days

The characteristics of the HAP hydrogel were observed using SEM, with the hydroxyapatite being clearly observed on the surface of hydrogel, and the distribution of HA into hydrogel shown to be quite constant and homogeneous (Figure 5A). In addition, SEM with EDS mapping analysis of the HAP hydrogel also showed that the presence of the ion density of the HA material was unidirectional with the SEM images (Figure 6A). These findings point to the biocompatibility of this material, because the distributed HA being constantly on the hydrogel surface may help the body fluid contact the graft material easily and enhance the effectiveness of bone augmentation.

After the HAP hydrogel was immersed in the SBF for 14 days, the SEM images showed new apatite formation on the hydrogel surface, and the amount of apatite was more noticeable compared with before immersion (Figure 5B). Furthermore, according to SEM with EDS mapping analysis, an increase in the ion density of the HA material was also exhibited on day 14 after immersion (Figure 6B).

### 3.3. Radiographic Findings and Grading of Post-Extraction Socket Preservation with HAP Hydrogel

Intraoral radiographs of a socket with HAP hydrogel placement are shown in Figure 7. When comparing intraoral radiographs of the socket before and immediately after HAP hydrogel placement, there was no distinctive difference in the socket radiodensity. Lamina dura, which is a radiopaque lined at the socket border, occurred on the day after extraction. In addition, the socket radiodensity was found to be more radiolucent when compared with the surrounding alveolar bone (Figure 7A).

In the 2nd week after extraction (Figure 7B), the gradual disappearance of lamina dura was observed, but it was still present at the coronal part of the socket. All post-extraction sockets in the 2nd week were classified as grade 2 sockets (slightly changed). Moreover, five out of seven of the sockets were considered grade 2 (ground glass appearance). The rest were considered grade 1 (changed). Further, radiographic findings showed an increase in the sockets’ radiodensity in this week.

Radiographic findings in the 4th week after extraction (Figure 7C) showed that the lamina dura had disappeared. It was difficult to distinguish the border of the socket wall from the surrounding alveolar bone. Five out of the seven sockets were classified as grade 3 (partly reduced) and two out of seven as grade 4 sockets (entirely absent). In addition, the radiopacity of most sockets was increased when compared with the previous follow-up times, especially in the apical area of the sockets. Three, three, and one of the seven post extraction sockets in this week were classified as grade 2, grade 3 (spicular appearance), and grade 4 (trabecular appearance), respectively.

The lamina dura of four of the seven sockets were entirely absent, and the rest were still partly reduced in week 8 (Figure 7D). In addition, the majority of the sockets were more radiopaque than at the previous follow-up time.

In the 12th week post-operation (Figure 7E), the lamina dura had disappeared entirely in all of the sockets, and the radiodensity of the sockets was comparable to that of the adjacent alveolar bone (7/7).

The number of sockets following the radiographic criteria in each week of follow-up are shown in Figure 8. In Criteria A and Criteria B, there were statistical differences (Table 2) in the radiographic gradings between the day of operation (Wk 0) and the 2nd, 4th, 8th and 12th weeks after operation.

### 3.4. Bone Height Analysis of Post Extraction Socket Preservation with HAP Hydrogel

The height of alveolar bone regeneration was measured to assess the effectiveness of the HAP hydrogel in the 2nd, 4th, 8th, and 12th weeks after post-extraction socket preservation. A paired *t*-test was used to analyze the differences between the day of the operation and the weeks of follow-up. The mean new alveolar bone height (Av.Bh) for each week is shown in Table 3. The differences in bone height were higher in the 2nd week (*p* = 0.045), 4th week (*p* = 0.045), 8th week (*p* = 0.019), and 12th week (*p* = 0.013) compared to the day of the operation (Figure 9a).

### 3.5. Bone Regeneration Analysis of Post-Extraction Socket Preservation with HAP Hydrogel

Bone intensity, measured using the image J program, was indicated as the radiodensity of alveolar bone within the socket using quantitative values. The mean bone intensity of the socket (Av.BI) is described in Table 3. The characteristics of bone intensity progression are related to increases in postoperative time. These values were analyzed using a paired *t*-test to assess bone regeneration between the days of the operation and different follow-up weeks. For bone regeneration analysis, the bone intensity of the post extraction sockets in the 2nd week (*p* = 0.039), 4th week (*p* = 0.009), 8th week (*p* = 0.01), and 12th week (*p* = 0.004) were higher than on the day of the operation (Figure 9b).

**Table 3 vetsci-09-00007-t003:** The mean and standard deviation of new alveolar bone height (Av.Bh) and alveolar bone intensity (Av.Bi) of post-extraction socket preservation in 2nd, 4th, 8th and 12th weeks.

Parameter	2nd Weeks	4th Weeks	8th Weeks	12th Weeks
Av.Bh (mm)	3.61 ± 1.47	4.14 ± 1.06	4.90 ± 1.51	5.92 ± 2.30
Av.BI	81.68 ± 12.70	85.91 ± 10.83	91.74 ± 5.63	94.77 ± 6.41

AvBh. Units of mm, data are presented as mean ± SD.

## 4. Discussion

This study is the first to demonstrate the effectiveness of a nanohydroxyapatite-based hydrogel (HAP hydrogel) in promoting alveolar bone regeneration in the post-extraction sockets of dogs with naturally occurring periodontitis. Moreover, this study described the post-extraction socket preservation technique in clinical dogs with periodontal disease. Through alveolar bone regeneration analysis, the post-extraction sockets of the recruited dogs showed socket alterations 2, 4, 8 and 12 weeks post-operation. Moreover, the homogeneous appearance of apatite formation on the material surface after simulated body fluid immersion for 14 days implies the potential of biocompatibility and bioactivity for post-extraction socket preservation. Consequently, these findings can shed light on a technique for reducing the risk of pathological mandibular fracture after dental extraction in dogs with moderate and severe grades of periodontitis. However, further investigation of the efficacy of the HAP hydrogel for post-extraction socket preservation in clinically normal dogs and dogs with periodontitis should be performed. In addition, the appropriate dosage of nanohydroxyapatite within the hydrogel should be varied.

In general, hydroxyapatite (HA) is the inorganic component of the natural bone composition. These characteristics lead hydroxyapatite to exhibit more biocompatibility than other calcium phosphate substitutes [16]. Although hydroxyapatite has osteoconductive properties and can improve bone formation by providing a scaffold for bone tissue defects, cell adhesion, and angiogenesis during the bone healing process, combining HA with a polymeric scaffold can provide greater efficacy in bone regeneration augmentation [1,39]. Moreover, the limitations of hydroxyapatite, such as its hardness, fragility, being difficult to organize in post-extraction sockets, and too high density for cell infiltration can be diminished by combination with another material [40]. In the current study, a polyacrylamide-based hydrogel (PAM), which is known as a synthetic hydrogel, was combined with hydroxyapatite for post-extraction preservation and for its potential osteoconductive benefits. Because the PAM exhibits hydrophilic properties with a three-dimensional scaffold, these properties support a suitable environment for bone tissue regeneration, including osteoblast infiltration, cell adhesion, and body fluid retention. Moreover, the advantages of the PAM include the strength of its network, and its toughness, degradability, and lower cytotoxicity, which are interesting options for application in the body [41,42].

The alveolar process is referred to as the alveolar socket, which is a part of the alveolar bone, and the main role of this structure is to support the tooth root. There are four parts of the alveolar socket structure, including the periosteum and dense compact bone, and cancellous bone as a general bone structure. However, the additional layer of alveolar bone is called the cribriform plate (referred to as lamina dura), which lines the alveolar sockets. Radiographically, lamina dura is the radiopacity structure that lines the border of the socket, and the appearance of lamina dura can imply information about teeth health. Moreover, alterations to this structure have been used to indicate the bone healing process and pathological changes in radiographic findings [43]. In the present study, although tooth extraction was performed, this structure still appeared on the day of the operation. However, Highlight’s study [36] suggested that the fading of lamina dura is related to the time after the operation and can be used to evaluate the rate of the socket healing process. The radiographic findings in this study demonstrated that post-extraction sockets treated with the hydroxyapatite-based hydrogel showed significant differences in lamina dura alteration when comparisons between the operation day and the different weeks of follow-up were made, which accords with Gupta’s study [44]. This study reported that the sockets were treated with chitosan for socket healing augmentation, and they showed better radiographic and complete disappearance of lamina dura than the control side at the 3 month follow-up.

Normally, the process of alveolar socket healing has been initiated once the dental extraction was performed [45,46], and the phase of bone healing in this area has been the same as the other bone healing processes, including the hematoma inflammation phase, repair phase, and remodeling phase. The evaluation of alveolar socket radiodensity under radiography is one of the common techniques used to practically assess alveolar bone healing in the clinical field [37,47]. A study in human dentistry reported that the radiodensity of new bone formation in the socket began to increase at 38 days, and increased until the radiodensity was equal to the bone surrounding the socket at the 105th day or 3 months after extraction [48]. On the other hand, the timing of alveolar socket healing under radiography in dogs has not been completely reported. In a previous study on socket healing in dogs, the socket radiodensity without material placement under extraoral radiography showed no differences between the day of operation and the 3 week follow-up [38]. Together with Discepoli’s study [49], this previous study described that the alveolar sockets in dogs after extraction at 3 months had not completely healed. These findings may imply that the phase of spontaneous socket healing using radiography at 3 weeks after the operation may not be detectable, and socket healing at 3 months after the operation is in progress. In this study, intraoral radiography was applied to assess radiodensity changes in the sockets treated with the HAP hydrogel each week after the operation. The characteristic of alveolar bone formation is one of the radiographic criteria which was adopted by Kattimani et al., 2014. [50] to assess the effectiveness of this material for post-extraction preservation.

The current study reported that radiographic and mean alveolar bone intensity values showed significant changes when comparisons were made between the day of operation and 2, 4, 8 and 12 weeks after the operation. Moreover, this study found that the sockets treated with the HAP hydrogel showed significant radiodensity changes at 2 weeks after the operation, and all of the sockets showed the same radiodensity for surrounding alveolar bone at the 12 week follow-up. These findings agree with previous studies on hydroxyapatite-composited material [20,32,40,51,52,53]. Therefore, these findings suggest that the HAP hydrogel has a positive osteogenic effect and reduces the duration of socket healing compared with the duration of socket healing without material placement [38,49]. Interestingly, Araujo’s study [54] reported that an alveolar socket with material placement showed newly formed bone only at the apical part of the socket, where the material did not persist. This phenomenon may happen because the area without material permits the normal bone healing process, i.e., blood clot accumulation and granulation tissue were replaced by a provisional matrix [31] and were not interfered with by foreign material. In contrast to the current study, alveolar socket healing was observed in the whole part of the socket on the last day of follow-up. Our study applied hydrogel as a scaffold for hydroxyapatite deposition, which can retain biological fluid, swell until it fulfills the socket, and degrade, which are the reasons why hydrogel is more biocompatible and does not interfere with the alveolar socket healing process.

In general, the empty alveolar socket can cause spontaneous alveolar bone resorption [11,12] Moreover, this phenomenon proceeds slowly throughout a lifetime. The cause of alveolar bone resorption is still unknown. However, decreased blood supply, atrophy disuse, and localized inflammation may be the factors that affect the rate of alveolar bone resorption. Although natural healing of the alveolar socket can happen after dental extraction, dogs with moderate and severe periodontitis usually show a few thicknesses between the apical area of the socket and the mandibular compact bone. Moreover, Kim’s study [55] suggested that the healing of extraction sites in dogs with periodontal and endodontic pathology may show more delayed healing than healthy extraction sites. These are the reasons why pathological mandibular fracture can occur in dogs with this disease. In the current study, bone height assessment was performed by measuring the distance from the highest part of the radiopaque area to the apical part of the socket under the extraoral radiograph in each week of follow-up. Interestingly, the bone height assessment in this study showed significant differences between the day of operation and the 2, 4, 8 and 12 weeks of follow-up. The progression of new alveolar bone regeneration increased following the post-operation follow-up. Moreover, this assessment was unidirectional with the statistical analysis of radiographic and mean alveolar bone intensity values. Consequently, this finding could be interpreted as suggesting that the HAP hydrogel can, in post-extraction sockets, enhance the alveolar socket’s apical to coronal alveolar healing.

Because hydroxyapatite can promote bone healing via osteoconductive properties, which act as scaffolds for cell infiltration and neovascularization [56,57], the evaluation of graft material distribution in a polymeric scaffold before use in clinical studies should be performed to assess the biocompatibility of the HAP hydrogel material. Scanning electron microscopy and energy-dispersive X-ray spectroscopy (EDS) are widely used to illuminate the characteristics of surface materials [58]. In the current study, SEM and EDS mapping analysis was used, and this showed that the spread of HA into the hydrogel showed a quite homogeneous distribution. Moreover, these findings lend support to the radiographic finding for alveolar socket regeneration that the entire socket after HAP hydrogel placement showed radiopacity on the last day of follow-up. In addition, the bioactivity of the HAP hydrogel material was estimated by immersing the material in a simulated body fluid (SBF) for 14 days and analyzing the new apatite formation under SEM and EDS. The formation of apatite on the material surface can indicate the ability of the implanted material to form interfacial bonds with tissue at the grafted area when in contact with the physiological fluid. Moreover, this technique can provide details about the bioactivity of the HAP hydrogel for bone healing augmentation before implantation and reduce the number of animal experiment samples needed for a study [34,59]. Our findings demonstrated that SEM images of the HAP hydrogel surface after immersion showed more apatite formation compared with before immersion. In addition, the SEM with EDS mapping analysis results also suggest that apatite was formed with constant distribution on its surface. Consequently, the HAP hydrogel has the potential to be used a biomaterial for post-extraction socket preservation in clinical use.

Intraoral radiography (IOR) is widely used to illustrate the structure of teeth, the periodontal apparatus, and alveolar bone by focusing on the oral cavity. In this study, this technique was adopted to assess alveolar bone regeneration with the HAP hydrogel in post-extraction sockets, because IOR is an effective method that is economical, easy to access, and fast to process, and can be performed during the surgical procedure [60]. Although cone beam computed tomography (CBT), an advanced imaging technique, may provide more accurate results in detecting bone radiopacity and tooth structure, CBT is not commonly available in animal hospitals and animal clinics, and the cost of this technique is quite higher than IOR. Moreover, alterations in the radiodensity of post-extraction sockets under IOR are adequate to indicate that the healing process is occurring. This study showed that the intraoral radiographs of post-extraction sockets with HAP hydrogel placement had increased radiopacity following weeks of follow-up. Therefore, it can be assumed that the HAP hydrogel led to positively enhanced alveolar bone regeneration in the sockets of the dogs with periodontitis.

Although the effectiveness of the HAP hydrogel for post-extraction socket preservation was investigated in this study, some unexplored variables of the oral and socket environment that may affect socket healing were not assessed. According to previous studies, prosthetic implants, such as prosthetic joints [61] and dental implants [62], may experience bacterial infection at implanted sites due to the adherence of bacterial biofilm on the surface of the implants post-operation [63]. Moreover, Kim’s study [55] suggested that the infected socket was characterized by vascular supply atrophy, which may affect the bone remodeling process. Consequently, postoperative ancillary therapies or anti-infective materials were suggested to reduce bacterial colonization in the oral cavity or implants [64,65]. To investigate the factors that can affect the HAP hydrogel’s potential for enhancing alveolar bone regeneration, the relationship between ancillary therapies that influence the oral environment and the efficacy of the HAP hydrogel for post-extraction socket preservation should be considered in future clinical trials.

One limitation of this study was that, while it was concluded that the HAP hydrogel can effectively enhance alveolar bone regeneration using radiographic findings, bone height measurement, and bone regeneration analysis, another bone healing assessment method, a bone tissue biopsy in the grafted area, is still essential to ensure the progression of socket healing processes. However, this technique was not performed in the present study, because dogs with naturally occurring periodontitis may not be good candidates for bone biopsies. These dogs have extensive alveolar bone destruction due to moderate to severe periodontitis. Moreover, bone biopsy techniques may be more traumatic and destructive to the intact alveolar bone. Thus, for further investigation, assessing the alveolar healing process after HAP hydrogel placement using bone tissue biopsies may be suggested to be performed for dogs with clinical periodontitis (mild periodontitis) or normal clinical dogs.

## 5. Conclusions

Radiographic findings, bone height measurement, and bone regeneration analysis all contributed to the conclusion that the HAP hydrogel was effective for post-extraction socket preservation in dogs with periodontitis. Moreover, scanning electron microscopy and energy-dispersive X-ray spectroscopy (EDS) confirmed that this material has the potential to promote new bone regeneration, as new apatite formation on the material surface in a simulated body fluid (SBF) after 14 days was observed, as well as the satisfying distribution of hydroxyapatite into the hydrogel surface. Our findings suggest that the HAP hydrogel has the ability to be an osteoconductive material that acts as a scaffold for promoting cell infiltration and neovascularization in the alveolar bone regeneration process, and can be applied for post-extraction socket preservation in dogs, especially in dogs with moderate and severe periodontitis. Further studies will be necessary to investigate the efficacy of this nanohydroxyapatite-based hydrogel in clinically normal dogs and dogs with periodontitis. Moreover, the dosage of hydroxyapatite within the hydrogel, which can provide an increased ability to promote alveolar bone healing, should be varied. In addition, the influence of ancillary therapies that affect the oral environment on the efficacy of the HAP hydrogel for post-extraction socket preservation should be investigated. To obtain the most efficacious results, this hydroxyapatite-based hydrogel material can be applied as a biomedical material in post-extraction socket preservation in dogs that suffer from periodontal disease in the veterinary dentistry field.

## Figures and Tables

**Figure 1 vetsci-09-00007-f001:**
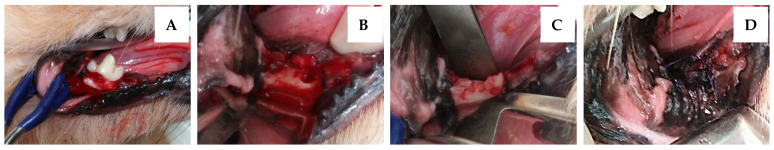
Steps for nanohydroxyapatite-based hydrogel (HAP hydrogel) placement in alveolar socket. (**A**) An open dental extraction technique was performed on the unilateral mandibular first molar tooth (309 or 409). (**B**) Post-extraction sockets of the mesial root and distal root were noted as tested sockets for post-extraction preservation. (**C**) Each post-extraction socket was filled with HAP hydrogel. (**D**) Mucoperiosteal flaps were used for the purpose of closing sockets after HAP hydrogel placement.

**Figure 2 vetsci-09-00007-f002:**
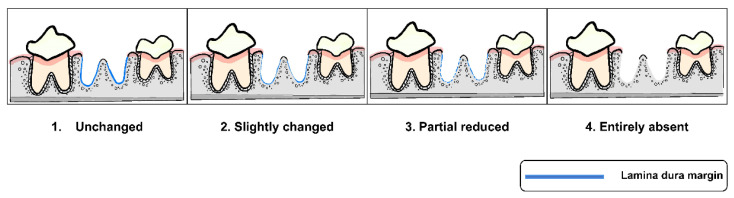
Characteristics of surgical site margin of post-extraction sockets in four grades (Criteria A).

**Figure 3 vetsci-09-00007-f003:**
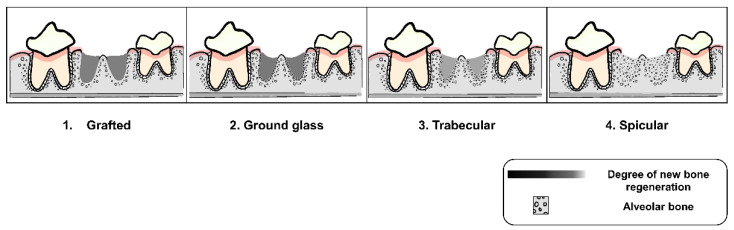
Characteristics of alveolar bone formation of post-extraction sockets in four grades (Criteria B).

**Figure 4 vetsci-09-00007-f004:**
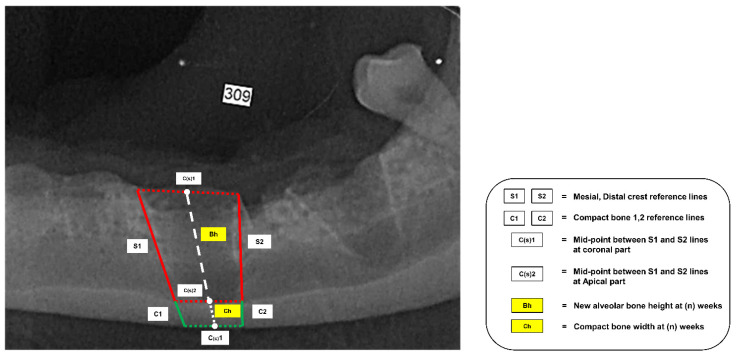
Intra-oral radiograph schematic with reference lines for bone height measurement in post-extraction socket using Vet Exam program.

**Figure 5 vetsci-09-00007-f005:**
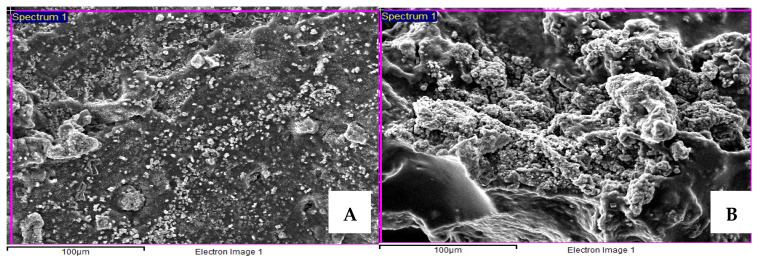
The scanning electron microscopy (SEM) images of nanohydroxyapatite-based hydrogel before and after immersion with simulated body fluid (SBF) for 14 days. (**A**) The image shows the distribution of hydroxyapatite composition within hydrogel. (**B**) The image shows new apatite formation at day 14 of SBF immersion.

**Figure 6 vetsci-09-00007-f006:**
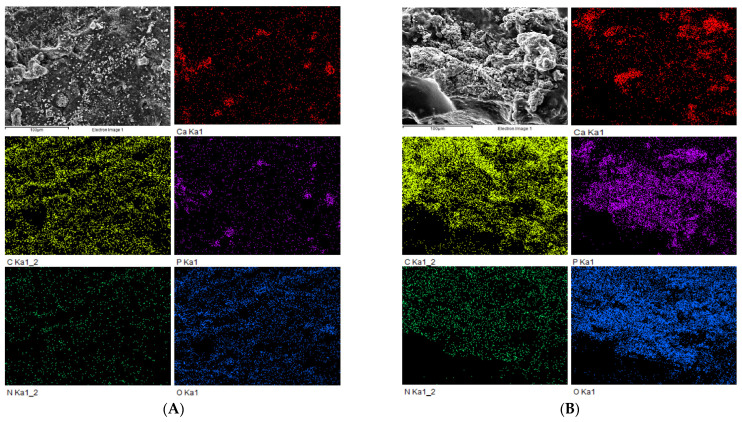
The scanning electron microscopy images (SEM) with energy-dispersive X-ray spectroscopy (EDS) mapping analysis, which were used to evaluate the distribution of hydroxyapatite-based hydrogel before (**A**) and after immersion (**B**) with simulated body fluid (SBF) for 14 days.

**Figure 7 vetsci-09-00007-f007:**
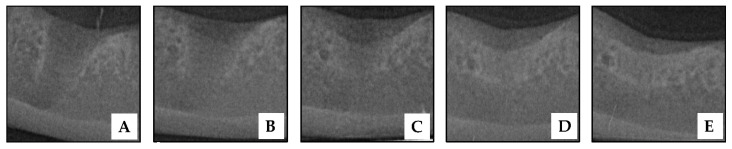
Intraoral radiographs of first mandibular molar socket after nanohydroxyapatite-based hydrogel placement in different weeks of postoperative follow-up including the day of operation (**A**), 2nd week (**B**), 4th week (**C**), 8th week (**D**), 12th week (**E**) after operation.

**Figure 8 vetsci-09-00007-f008:**
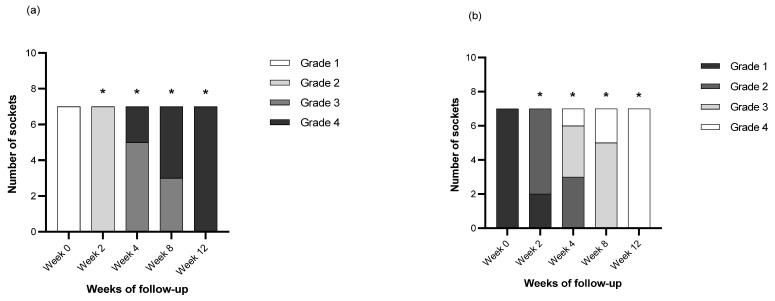
The number of sockets following characteristics of surgical site margin (**a**) and characteristic of new alveolar bone formation (**b**) in each week of follow-up are represented with bar plots. An asterisk on top of a bar indicates a statistically significant difference at a *p*-value of 0.05.

**Figure 9 vetsci-09-00007-f009:**
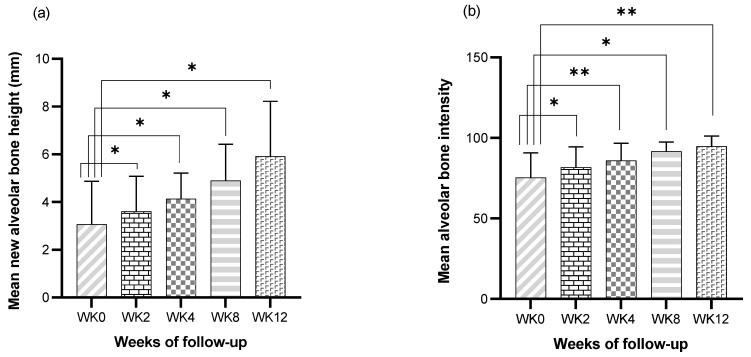
Comparison of alveolar bone regeneration parameters including new alveolar bone height (**a**) and alveolar bone intensity (**b**) among the day of operation and 2nd (WK2), 4th (WK4), 8th (WK8) and 12th weeks (WK12) after operation. An asterisk and two asterisks on top of a bar indicate a statistically significant difference at a *p*-value of 0.05 and a *p*-value of 0.01, respectively.

**Table 1 vetsci-09-00007-t001:** Signalments including age, sex, breed, body weight, periodontal stages and concurrent diseases of five dogs.

Age (Years)	11 (8–15)
Sex	4 Male, 1 Female
	(2 intact male, 2 neuter male, 1 neuter female)
Breeds	Poodle (2/5)
	Shih Tzu (1/5)
	Pomeranian (1/5)
	Chihuahua (1/5)
Body weight	4.2 (3.55–9.8)
Periodontal stages	Grade 1 (0/5)
	Grade 2 (0/5)
	Grade 3 (2/5)
	Grade 4 (3/5)
Concurrent Diseases	Hepatobiliary problem (1/5)
	Myxomatous mitral valve degeneration (1/5)
	Unremarkable (3/5)

Data are presented as median (range) or total number values.

**Table 2 vetsci-09-00007-t002:** Comparison of radiographs following changes in lamina dura around socket (Criteria A) and characteristic of new alveolar bone formation following changes in radiodensity within socket (Criteria B) between the day of operation and 2nd, 4th, 8th and 12th postoperative weeks.

Follow-Up Weeks	*p*-Value	
	Criteria A	Criteria B
Week 2	0.010	0.036
Week 4	0.017	0.020
Week 8	0.019	0.017
Week 12	0.010	0.010

## Data Availability

Not applicable.

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
