# Peer review of "Effectiveness of a Nanohydroxyapatite-Based Hydrogel on Alveolar Bone Regeneration in Post-Extraction Sockets of Dogs with Naturally Occurring Periodontitis"

_vetsci, 2021, doi:10.3390/vetsci9010007_

Round 1

Reviewer 1 Report

  The revised manuscript is about a new technique for alveolar bone regeneration after molar extraction in dogs affected by periodontitis.  Material and methods are described in detail.  Ethical standards are maintained in the experimental management. Results are clearly exposed with informative tables and quality figures. Supplementary tables are interesting since they provide important details.  The results have been adequately discussed. Reference list is updated: 35/59 (59 %%) references refer to the last 10 years and 19/59 (32%) references are even more recent, from the last 5 years.

English language must be revised. Please correct the punctuation marks. Some subordinate phrases lack a verb; also, subordinate phrases must be separated from the main one by commas or semi colon Capital letters must be used after a period and not after a comma

Minor specific points:

 Line 35:  the SEM with EDS mapping of HAP-hydrogel…  Abbreviations must be defined the first time they appear in the text

Lines 79-80:  Please explain the main differences between hydroxyapatite and Nano-hydroxyapatite

Lines 253-258: these sentences repeat part of what was said in animal material. Please, delete.

 Lines 259-260: It is said “Unilateral 1st mandibular molar extraction was done and there are ten post extraction sockets”. Only one 1st mandibular molar was extracted in each of the five dogs? Then, it is hard to understand how ten post extraction sockets were considered.  Please, explain,

 Line 260: It is said “fives dog” ; it must be said “five dogs”.

Reviewer 2 Report

Dear Authors,
I have read the manuscript with interest and some questions raised. Enlisted please find my comments.
Overall. General English grammar revision (Minor spelling errors).
Key words. “veterinary dentistry” and “veterinary periodontology” could be added in my opinion.
Abstract. Please add the names of the statistical tests in this section.
Introduction. Authors stated “PD is the result of the inflammatory response of the periodontal apparatus to the dental plaque on the tooth surface ”. Please add a reference for this statement.
Materials and Methods. Authors stated “A total of five small breed dogs with age > 3 years regardless of sex and bodyweight that was brought as an outpatient at the dental clinic, Small Animal Teaching Hospital, ”. Please add if and how sample size calculation has been performed.
Materials and Methods. Authors stated “In addition, scanning Electron Microscopy (SEM) with Energy dispersive x-ray spectroscopy (EDS) analysis was performed using JEOL JSM-6335 F electron microscope to evaluate the distribution of hydroxyapatite and apatite formation on hydrogel’s surface before and after immersion with SBF for 14 161 days ”. Please add details about microscope: manufacturer, City and State. 
Materials and Methods. Authors stated “Before the initial operation, mouth washing with 0.12% chlorhexidine gluconate”. Please add details about commercial name, manufacturer, City and State. 
Figure 1. Please enlarge the figures in order to see better the details. If necessary, please split in 4 diffferent images
Materials and Methods. Authors stated “Genoray Port-X II Portable Dental X-ray system was used for intraoral radiography”. Please add details about commercial name, manufacturer, City and State. 
Figure 4. Please enlarge the figures in order to see better the details. Legend is not visible. Picture details are not visible.
Materials and Methods. Authors stated “Statistical analysis was analyzed with R program (The R Foundation) ”. Please add details about software used: version, Manufacturer, City and State.
Materials and Methods. Authors stated “Data from the statistical analysis were represented as graphs generated by GraphPad”. Please add details about software used: version, Manufacturer, City and State.
Figure 8. Please enlarge the figures in order to see better the details. If necessary, please split in 2 diffferent images.
Discussion. Provide a general interpretation of the results in the context of other evidence, and implications for future research. Some discussion should be added concerning possible ancillary helpful therapies. It could be stated that “Some unexplored variables can have a significant influence on oral environment. The use of probiotics (Butera A, Gallo S, Maiorani C, Preda C, Chiesa A, Esposito F, Pascadopoli M, Scribante A. Management of Gingival Bleeding in Periodontal Patients with Domiciliary Use of Toothpastes Containing Hyaluronic Acid, Lactoferrin, or Paraprobiotics: A Randomized Controlled Clinical Trial. Applied Sciences. 2021; 11(18):8586. ) and natural compounds (Chitosan and Hydroxyapatite Based Biomaterials to Circumvent Periprosthetic Joint Infections. Costa-Pinto AR, Lemos AL, Tavaria FK, Pintado M. Materials (Basel). 2021 Feb 8;14(4):804.)can modify Clinical and Microbiological Parameters in periodontal patients, they could have an effect also in socket healing. All these variables should be considered in future clinical trials.”.
Discussion. Please add a paragraph concerning the treatment alternatives.
Discussion. Please add a paragraph showing the limitations of the present report.
Discussion. 
References. Some references are quite old (1941;2000). If possible, please switch with some more modern research. Some recent studies have been suggested in the sections above.

Round 2

Reviewer 2 Report

Good job